# Optimization of SPIO Injection for Sentinel Lymph Node Dissection in a Rat Model

**DOI:** 10.3390/cancers13195031

**Published:** 2021-10-08

**Authors:** Mirjam C. L. Peek, Kohei Saeki, Kaichi Ohashi, Shinichi Chikaki, Rose Baker, Takayuki Nakagawa, Moriaki Kusakabe, Michael Douek, Masaki Sekino

**Affiliations:** 1Division of Cancer Studies, King’s College London, London SE1 9RT, UK; mirjam_peek@hotmail.com (M.C.L.P.); michael.douek@kcl.ac.uk (M.D.); 2Graduate School of Agricultural and Life Sciences, University of Tokyo, Tokyo 113-8657, Japan; anakaga@g.ecc.u-tokyo.ac.jp (T.N.); amkusa@mail.ecc.u-tokyo.ac.jp (M.K.); 3Department of Electrical Engineering and Information Systems, Graduate School of Engineering, University of Tokyo, Tokyo 113-8654, Japan; ka1.ohashi.702@gmail.com (K.O.); chikaki@g.ecc.u-tokyo.ac.jp (S.C.); sekino@g.ecc.u-tokyo.ac.jp (M.S.); 4Department of Statistics, University of Salford, Salford M5 4WU, UK; r.d.baker@salford.ac.uk

**Keywords:** sentinel lymph node dissection, sentinel lymph node, super paramagnetic iron-oxide particles, rat model, magnetic technique, magnetic tracer

## Abstract

**Simple Summary:**

In this study, the following injection characteristics were evaluated to optimize magnetic tracer uptake in the sentinel lymph nodes (SLN) in a rat hindleg model: (a) iron dose, (b) effect of dilution, (c) effect of injecting at different time courses and (d) effect of massaging the injection site. In conclusion, injection dose and time were primary factors for the SLN iron uptake. The result from this study will provide a background for magnetic procedures.

**Abstract:**

The magnetic technique, consisting of a magnetic tracer and a handheld magnetometer, is a promising alternative technique for sentinel lymph node dissection (SLND) and was shown to be non-inferior to the standard technique in terms of identification rates. In this study, injection characteristics (iron dose, dilution, time course and massaging) were evaluated to optimize magnetic tracer uptake in the sentinel lymph nodes (SLN) in a rat hindleg model. 202 successful SLNDs were performed. Iron uptake in the SLN is proportional (10% utilization rate) to the injection dose between 20 and 200 μg, showing a plateau uptake of 80 μg in the SLN around 1000 μg injection. Linear regression showed that time had a higher impact than dilution, on the SLN iron uptake. Massaging showed no significant change in iron uptake. The amount of residual iron at the injection site was also proportional to the injection dose without any plateau. Time was a significant factor for wash-out of residual iron. From these results, preoperative injection may be advantageous for SLN detection as well as reduction in residual iron at the injection site by potential decrease in required injection dose.

## 1. Introduction

Sentinel lymph node dissection (SLND), also referred to as sentinel lymph node biopsy, is the standard of care for clinically and radiologically node-negative breast cancer patients to stage the axilla and determine if cancer has spread to regional lymph nodes [1,2]. The current standard technique consists of a radioisotope and blue dye which are injected subcutaneously in the breast on the day of surgery [3]. However, this technique has some drawbacks including strict regulations regarding the use of radioisotopes and complications related to the use of blue dye [4]. The magnetic technique consisting of magnetic nanoparticles and a handheld magnetometer is a promising alternative technique for SLND. This magnetic technique was shown to be non-inferior to the standard technique in several trials and meta-analyses [4,5,6,7,8,9,10,11,12,13,14,15,16,17]. There are different magnetic nanoparticles on the market but the most commonly used are Sienna+ (Endomag Ltd., Cambridge, UK), Magtrace (former SiennaXP, Endomag Ltd., Cambridge, UK) and Resovist (Schering AG, Berlin, Germany) with iron concentrations of 28.0, 28.0 and 27.9 mg/mL, respectively [18,19].

Recently, it was discovered that residual magnetic nanoparticles at the injection site can lead to susceptibility artifacts on breast MRI [20]. This could be an issue in patients who need post-operative MRI such as BRCA carriers, patients who have SLND prior to primary treatment or patients whose tumors are not visible on mammography. Therefore, it is important to reduce the volume of magnetic nanoparticles injected to optimize this technique.

Most clinical trials used a magnetic nanoparticle volume of 2.0 mL diluted with saline to 5.0 mL, injected on the morning of or just prior to surgery followed by a five-minute massage of the injection site [4,5,6,7,8,9,10]. Hersi et al. [16] performed a study comparing injections at lower iron doses and different injecting timings, and Rubio et al. [19] compared Magtrace injections of 1.0, 1.5 and 2.0 mL. Recently, studies have successfully employed injection days or even weeks prior to surgery, revealing that preoperative injection is also feasible [21,22,23]. Knowledge from the previous studies indicates that iron dose, time course of injection, dilution, and massaging could affect iron uptake by the sentinel lymph nodes (SLN). Optimization of these factors is important in order to reduce the residual magnetic nanoparticles left at the injection site.

In this study, injection characteristics were evaluated to optimize magnetic tracer uptake in the SLN in a rat model. We performed multiple experiments to determine (a) the optimum iron dose, (b) the effect of dilution, (c) the effect of injecting at different time courses and (d) the effect of massaging the injection site.

## 2. Materials and Methods

Animal studies were approved by the local Ethics Board (Accession Number P15–124) and the experiments were performed between December 2015 and March 2016 at the University of Tokyo according to the guidelines of the institution and ensured humane care of animals. Authors adhered to the ARRIVE guidelines [24] (Appendix A). No sample size calculation was performed and the number of animals in each group (*n* = 4–5) was determined according to the previous publication [25]. Female Sprague Dawley rats of ten weeks old (Nihon SLC, Shizuoka, Japan) (approximately 200 g) were randomly allocated to experiments. A flow diagram of the entire study can be found in Appendix A.

### 2.1. Dose Increase Experiment

A set volume of 100 μL was injected bilaterally in the subcutis between the second and third digits of the hind legs (Figure 1a). Resovist was injected manually on the right side and saline (control) on the left side. The magnetic tracer was diluted with saline to set the iron dose at 2, 10, 20, 40, 100, 200, 1000, 2000, 2790 and 4000 μg per 100 μL. In order to obtain the 4000 μg of iron per 100 μL of solution, two 0.8 mL vials of Resovist were centrifuged at 20 degrees Celsius for twelve hours with a relative centrifugal field of 20,000 G. After centrifugation, 0.4 mL of aqueous supernatant was carefully and aseptically discarded, and residual liquid and sediment were mixed thoroughly using a vortex mixer. A small sample (2 μL) was evaluated using a superconducting quantum interference device (SQUID) and an iron concentration of 1.4 times the standard Resovist concentration, 40.0 mg/mL, was found. All samples were evaluated for aggregation of nanoparticles using the fiber-optics particle analyzer with autosampler (FPAR-1000AS, Otsuka Electronics Co. Ltd., Osaka, Japan).

Prior to injection, all rats were marked, shaved and anesthetized using a mixture of 2–3% Isoflurane (Wako Pure Chemical Industries, Osaka, Japan) and 300 mL/min air via an automatic delivery system (Isoflurane Vaporizer SN-487, Shinano Seisakusho, Tokyo Japan), first using an induction chamber and, after the rats were anesthetized properly, through a mouthpiece during the procedure. Each iron dose was injected in five rats except for 2000 μg, which was injected in two groups of five rats by two researchers (M. Peek and K. Saeki). This was done to confirm reproducibility of the results between researchers, making the total number of rats used for this experiment 55. After injection, the rats were placed back in their cages for recovery.

After 24 h, the rats were anesthetized and euthanized by cervical dislocation. In rats, the popliteal lymph nodes are the primary and dominant SLNs draining the distal hindleg including the injection site. Following euthanasia, SLND of the popliteal nodes was performed bilaterally in a prone position. The caudal skin of the stifle joint was incised, and the popliteal lymph nodes were anatomically located within the thigh muscles and dissected (Figure 1a). Collected lymph nodes were placed in formalin.

In selected animals from the 1000 (*n* = 2), 2000 (*n* = 4) and 4000 (*n* = 5) μg iron injection groups, abdominal lymph nodes were also harvested as secondary lymph nodes: Briefly, the animals were placed in dorsal recumbency, and celiotomy was performed. At the caudal furcation of the descending aorta, iliac lymph nodes were identified and resected.

For all animals, the hindlegs were amputated bilaterally at the tarsal joints, weighed, and placed in a drying oven for 48 h at 80 degrees Celsius. The dry weight of the distal legs was determined prior to powdering them using TissueLyser (20 Hz, 2 min; Qiagen, Hilden, Germany), for further analysis. The excised sentinel lymph nodes (SLN) and the powdered rat legs were analyzed by SQUID to determine the amount of iron within the samples. The SLN uptake rate was subsequently calculated as (Iron accumulation in SLN)/(Injected iron dose) × 100.

### 2.2. Dilution and Time-Course Experiment

Resovist equivalent to 200 μg iron (7.17 μL) was diluted with saline two-, five-, and ten-fold, which resulted in final volumes of 14.34 μL, 35.84 μL, and 71.68 μL, respectively. All samples were evaluated for aggregation of nanoparticles using the FPAR-1000AS. A set iron dose of 200 μg was chosen, as any higher iron dose would already reach the plateau level uptake, thereby enabling the evaluation of dilution and time on the iron uptake.

All rats were marked, shaved and anesthetized using the same method as described above. Indicated Resovist solutions were injected bilaterally in the subcutis between the second and third digits of the hind legs, using an automated injection pump (MCIP-Jr, Minato Concept, Tokyo, Japan). The injection duration was set at 15 s independent of differences in injection volumes. During injection, the minimum and maximum pressures were recorded. SLND was performed after 10 and 30 min and 1, 6 and 24 h. Each sampling was performed bilaterally on two rats, giving four datasets per harvesting time point per dilution, a total of 80 datasets in 40 rats.

After injection, rats were placed back in their cages for recovery and SLND was performed after the indicated time frames. All rats were anesthetized and euthanized by cervical dislocation and bilateral SLND of the popliteal nodes was performed, as described for the dose increasing experiments.

As for the animals euthanized at 24 h after injection, abdominal nodes were excised in addition to the popliteal SLNs. The excised lymph nodes were placed in formalin and analyzed with SQUID. The distal hindlegs of the rats were processed as described above and analyzed with SQUID.

### 2.3. Massage Experiment

The rats were anesthetized as described above. Resovist was diluted 10 times with saline, and 71.7 μL of the solution (equivalent to 200 μg iron) was manually injected bilaterally in five rats; on the right side, this was followed by a five-minute massage of the injection site. The massage was manually performed with a one-second hold and one-second release cycle on the subcutaneous dome initiated by the injection. Rats were placed back in their cages for recovery. After 30 min, the rats were anesthetized and euthanized by cervical dislocation and SLND of the popliteal nodes was performed, as described for the dose increasing experiments. Distal hindlegs were processed and both injection sites and SLNs were analyzed with SQUID, as described above.

### 2.4. MRI Experiments

Imaging was performed using a 7.0 T BioSpec high-field small animal MRI system (Bruker Biospin, Germany). T1-weighted (T1W) MRI images with FLASH sequence were acquired in axial orientation without fat suppression and with the following parameters: TR/TE = 892.3/5.4 ms; FOV = 60 × 60 mm; matrix = 256 × 256; slice thickness = 1.0 mm; inter-slice distance = 1.0 mm; FA = 40 degrees; isotropic in-plane resolution = 0.14 mm. The maximum diameter of the artifacts at the SLNs caused by magnetic nanoparticles was recorded.

MRI was performed in rats who were injected with 2, 20, 40, 100, 200 and 2000 μg of iron (five rats per group) during the iron increasing experiments, and two age-matched untreated rats (control). MRI was performed to evaluate the size of the artifacts at the SLNs caused by magnetic nanoparticles. The animals were euthanized 24 h after injection, immediately followed by MRI scanning and harvesting of the SLNs.

For a single rat, continuous MRI scans were performed to visualize the uptake of magnetic nanoparticles within the SLNs. The rat was anesthetized using an intravenous injection of alpha-chloralose (approximately 50 mg/kg/h, to effect), placed in a prone position and breathing was monitored whilst scanning. A bilateral injection was performed with neat (7.17 μL; left) and 10 times diluted (71.7 μL; right) Resovist, both equivalent to 200 μg iron. MRI was performed every three minutes for the first hour and every ten minutes for the subsequent four hours. After five hours, the rat was euthanized, and SLND was performed on the popliteal, groin and abdominal nodes. The nodes were placed in formalin, and the distal hindlegs were processed with SQUID, as described above.

### 2.5. SQUID Measurements

The magnetic moment of the magnetic tracer contained in the extracted SLNs and powdered distal hindlegs were measured using SQUID (MPMS-5S, Quantum Design Inc., San Diego, CA, USA). Each node was placed in a capsule in the middle of a plastic cylinder and placed in the machine. The SQUID consisted of a uniform measurement magnetic field with a range between −300 to +300 mT (−3000 to +3000 Oe) and a magnetic field detection coil to measure the change of interlinked magnetic flux.

The measurement region showed linear curves for living tissue due to the diamagnetic effect, and non-linear curves for living tissue containing the magnetic tracer due to the mixed diamagnetic and super-paramagnetic effects. To determine the magnetic moment of the magnetic tracer contained in the SLNs, the mixed signal was separated using the least squares method (Mathematica, Wolfram Research Inc., Champaign, IL, USA) and the non-linear curve was extracted.

### 2.6. Statistical Analysis

Nonparametric (two-sided) tests were used wherever possible. The correlation between iron accumulation in the SLNs and the injection sites, and the injected dose was calculated using the Spearman (rank-based) correlation. Statistical differences in iron uptake between massage and no-massage groups were assessed using the Mann–Whitney U-test. The Spearman correlation was also used to examine associations between MRI artifact size and iron uptake, dilution or time of injection. A parametric analysis had to be used to simultaneously regress iron accumulation in the SLNs on dilution, time, and the amount of iron at the injection site. This multiple regression is described later.

All statistical analysis and visualization were performed using IBM SPSS Statistics Version 23 (SPSS Inc., Chicago, IL, USA) and results were reported according to the SAMPL guideline [26]. No criteria for exclusion was set and all the collected data were included.

## 3. Results

SLND was performed in 202 procedures (101 rats) except for two control animals in MRI experiments, and at least one popliteal lymph node (hereafter, SLN) was obtained with each procedure (Figure 1a). The higher dose injection of Resovist resulted in apparent brown discoloration of the SLNs (Figure 1a). Abdominal nodes (hereafter, the secondary nodes) were harvested in eleven rats from the dose increase experiment and eight rats from the 24 h harvesting time point group of the dilution and time-course experiments. Furthermore, a total of 202 hindlegs were amputated at the tarsal joint, dried and powdered for SQUID analysis.

In the dose increase experiments, the fiber-optics particle analyzer showed no aggregation for samples of 1000 μg or higher. Aggregation was found for samples with an iron dose of 200 μg and 100 μg (98.9 ± 1.4 and 127.0 ± 15.9, respectively, a particle size higher than neat Resovist 2790 μg: 73.1 ± 4.09 nm). In the dilution and time course experiments, no aggregation was found for the two-times diluted sample. Aggregation was detected in the five- and 10-times diluted samples (80.0 ± 5.7 and 86.8 ± 0.71, respectively, a particle size higher than neat Resovist: 73.1 ± 4.09 nm).

### 3.1. Dose Increase Experiment

#### 3.1.1. Iron Uptake in the SLNs

SQUID analysis of the SLNs demonstrated a plateau uptake of iron in the SLNs of approximately 80 μg (Figure 1b, Appendix A). This plateau was reached by injecting an iron volume of 1000 μg or higher. Prior to this plateau, the amount of iron taken up by the SLNs increased with increasing amounts of iron injected, which resulted in an overall significant strong correlation between the injected iron dose and the amount of iron in the SLNs (Spearman’s *r_s_* (56) = 0.92, *p* < 0.001) (Figure 1b, Appendix A). By calculation, it was evident that the utilization rate (i.e., iron in node/iron injected) between injected iron volumes of 10–200 μg was approximately 10%, with the highest utilization rates at an injected iron volume of 20 μg (11.0%), compared to 100 μg (10.0%) and 200 μg (9.7%) (Figure 1c). The utilization rate dropped significantly to 6% upon injection of 1000 μg iron, compared to 20-μg injection (Figure 1c). For the 4000 μg injection, this ratio dropped to 2%. The secondary nodes contained similar amounts of iron to in the SLNs under the investigated conditions (1000-, 2790- and 4000-μg injections) (Figure 1d).

#### 3.1.2. Iron at the Injection Site

The amount of residual iron after 24 h at the injection site increased as the injected iron dose increased, with a statistically strong correlation (Spearman’s *r_s_* (56) = 0.96, *p* < 0.001, Figure 1b, Appendix A). Iron at the injection site did not show any plateau accumulation as observed with SLNs, which indicates that an excessive injection of iron only results in an increased residue of iron at the injection site (Figure 1b,c).

#### 3.1.3. Difference between Researchers Performing Experiments

Experiments with the 2000 μg iron group were repeated in two settings by different researchers (M. Peek and K. Saeki) to determine the inter-researcher variability. The median iron amounts in the SLNs were 87.5 μg (range, 57.9–106.0 μg; K.S.) and 73.6 μg (range, 61.7–139.0 μg; M.P.). The median iron amounts at the injection site were 1192.5 μg (range, 1101.5–1243.1 μg; K.S.) and 1316.4 μg (range 1170.8–1594.4 μg; M.P.). No statistical difference was found between the researchers about iron amounts in the SLNs (Mann–Whitney U = 12.0, *p* = 1.00) and iron amounts at the injection site (Mann–Whitney U = 6.0, *p* = 0.22), supporting experimental consistency. In the other experiments, the two researchers performed procedures interchangeably and were not distinguished.

### 3.2. Dilution and Time-Course Experiment

#### 3.2.1. Iron Uptake in the SLNs

SQUID analysis showed an increase in iron uptake in the SLNs over time (Figure 2a–c, Appendix A). An increase in iron uptake was seen six hours after injection (Figure 2c). The first hour showed no large difference in iron uptake. Although the effect of dilution was not apparent at most time points, the five- and ten-fold dilution group demonstrated an increased iron uptake after six hours, with little difference in iron uptake at 24 h (Figure 2b). Neat Resovist showed barely any uptake within the first hour after injection whilst all diluted volumes were actively taken up by the SLNs. The highest iron uptake was seen with the ten-fold diluted tracer after 10 min, 30 min, and 6 h and the lowest overall uptake was seen with neat Resovist. The uptake at 24 h was, however, very similar with all dilution volumes.

#### 3.2.2. Iron at the Injection Site

At the injection site, the amount of iron per dilution group showed no difference, however over time, the iron amount slowly tended to decrease in all dilution groups (Figure 2d–f, Appendix A).

#### 3.2.3. Statistical Analysis

To find the best single predictors of iron volume in the SLNs and at the injection site, Spearman correlations were calculated. Correlation coefficients of iron volume in the SLNs with dilution, time, and iron volume at the injection sites are 0.201 (*p* = 0.07), 0.810 (*p* < 0.001) and −0.624 (*p* < 0.001), respectively. This result showed that time is a strong predictor for iron volume in the SLNs with statistical significance. To study the uptake of iron in the SLNs, it was necessary to regress SLN iron accumulation on time, dilution, and iron concentration at the injection site. For this purpose, linear and squared terms of the predictors were used. In addition, the SLN iron concentration needed to be transformed. The Box–Cox transformation y = (x^α^ − 1)/α was used, which reduces to a log transformation as α→0. For the SLN regression, α had a 95% confidence interval of (0.3, 0.59). Hence, for simplicity, the square root of SLN iron volume was used in the regression. Detailed results are given in Appendix A and confirmed that dilution was not a significant predictor. Time was, however, and also time squared, with a negative coefficient, showing that iron accumulation increases sublinearly, i.e., it slows down with time. Turning to iron volume at the injection site, the correlation coefficients with dilution and time are 0.0108 (*p* = 0.92) and −0.554 (*p* < 0.001), respectively. Again, time is the best predictor, and dilution has no significant effect. For multiple regression, the iron volume at the injection site was fitted with α = 1. As a result, dilution was not significant, whereas again the decrease with time was significant, and also slowed down significantly with time. Full results are given in Appendix A. Consequently, time after injection is the most significant factor for both increasing iron accumulation in the SLNs and facilitating iron clearance at the injection site. Dilution does not have a significant effect on either.

Iron accumulation in the secondary nodes showed no difference between the different dilutions, 24 h after injection (Figure 2g). There was no apparent trend in injection pressure when compared between the different dilution groups, despite the varying injection volumes. In addition, no animal showed extremely high or low injection pressure, which would indicate inadequate injection (intradermal or intravenous injection) of SPIO. Averaged minimal pressure was 22.95 ± 5.78 mmHg and averaged maximal pressure was 97.34 ± 27.91 mmHg.

### 3.3. Massaging Experiment

Massaging of the injection site had no significant increasing effect on the amount of iron taken up by the SLNs (Mann–Whitney U = 5.0, *p* = 0.15) if the SLND was performed 30 min after injection of the magnetic tracer (Figure 3). The SLNs on the massage side contained a slightly higher amount of iron (median 3.25 μg, range 2.90–4.62 μg) compared to the untreated side (median 1.67 μg, range 1.30–4.28 μg). No statistical significance in iron at the injection site was found between the massage and no massage groups (Mann–Whitney U = 11.0, *p* = 0.84). The injected site on the massage side contained a median iron amount of 173.65 μg (range 108.95–338.96 μg), and that of the untreated side had a median amount of 174.14 μg (range 135.55–251.22 μg).

### 3.4. MRI Experiment

The SLNs of 30 rats from the dose increase experiment and two controls were successfully scanned with MRI immediately after euthanasia. On T1-weighed images, SPIO accumulation in the SLNs was observed as a signal reduction of the entire lymph node in the 2-μg group and as a spherical artifact in the other five groups (Figure 4a). Control groups showed no signal reduction on MRI.

The size of the artifact at the SLNs increased with the injected dose of iron (Spearman’s *r_s_* (30) = 0.94, *p* < 0.001, Figure 4b, Appendix A). In addition, a statistically significant correlation was found between the amount of iron in the SLNs determined by the SQUID and the artifact size (Spearman’s *r_s_* (30) = 0.97, *p* < 0.001, Figure 4c, Appendix A).

In the time-course MRI experiment, iron uptake in the SLNs was seen immediately after injection and a reduction in signal was immediately visible at both sides (Figure 4d). The artifact located at the SLN was on average 4.64 mm on the left (×1) and 4.84 mm on the right (×10) in maximum diameter, showing no major difference in artifact size between neat and 10-times diluted Resovist. No increase in artifact size over time was seen on both sides after reaching plateaus around 30–60 min (Figure 4e). The absolute amounts of iron in the left and the right SLNs were 5.48 and 8.70 μg after resection, respectively.

## 4. Discussion

The amount of iron in the SLN was proportional to the injection dose up to 200 μg. However, excessive injection of iron only causes an increase in residual iron at the injection site, as there was a plateau uptake of 80 μg in the SLNs. The total amount of iron at the injection site increased (with increasing injection dose) without any plateau. This suggests that each lymph node has a maximum capacity to accommodate magnetic particles. The maximum accommodation of iron in lymph nodes may depend on size of lymph node. Therefore, it may differ not only between different animal species, but also between individuals of the same species and the locations of the nodes in the body. Size of SPIO particles may matter as well. Injecting excessive iron would therefore not be of any additional benefit and would most likely cause larger MRI artifacts at the injection site. This indicates that it is important to determine the precise plateau point to optimize the injection volume and reduce the possibility of a susceptibility artifact. This study, by investigating wide range of SPIO injection volumes, revealed both a gradual increase in iron in the SLNs along with an increase in iron injection dose and the existence of a plateau in uptake. Proportional increase in iron in the SLNs was observed between 20 (0.1 mg Iron/BW in a 200 g rat) and 200 μg (1 mg Iron/kg BW) iron injection. For a person who weighs 60 kg, 2 mL of neat SPIO (either of Sienna+, Magtrace, or Resovist) injection is equal to 1 mg Iron/kg BW, approximately. Utilization rates were also low with injection of small iron volumes less than 10 μg. This could be caused by injected iron being trapped at the injection site such as phagocytosis by tissue macrophages, but further evaluation is required. Alternatively, this could be due to aggregation of the SPIO at lower iron concentrations, preventing uptake into the SLNs. This phenomenon, aggregation of the SPIO particles at low concentration, has not been identified so far. In most human clinical trials, SPIO agents were traditionally diluted at 2.5 times to facilitate SLN uptake. However, excessive dilution of SPIO may decrease iron uptake by the lymphatic system in human medicine as well.

During dilution experiments, longer time from injection increased iron uptake in the SLNs to a greater extent than increasing the dilution factor. This indicates that the lymphatic system may be able to take up more magnetic tracer if the injection is performed a day prior to surgery rather than injection just prior to surgery, as has already been shown in some studies [16,21,22,23]. For iron at the injection site, time is also a significant factor for washout. Dilution did not have a significant effect on either iron in the SLNs or at the injection site. Massaging may facilitate iron uptake by the SLNs, but the effect was small and not statistically significant in this study.

Although no human model was used for this study, it may be possible to suggest critical factors and conditions to optimize SPIO injection for breast cancer patients, together with the previous studies. The use of a magnetic tracer for SLND was first evaluated in a porcine model [18,25,27]. Anninga et al. [27] showed a significant correlation between magnetometer counts and iron in SLNs (*r* = 0.86, *p* < 0.01). Grading with both H&E and Perl’s staining showed a correlation with iron content (*p* = 0.001, *p* = 0.003) and magnetometer counts (*p* < 0.001, *p* = 0.004). Pouw et al. [18] evaluated three different magnetic tracers and showed that Sienna+ had the highest number of detected transcutaneous hotspots and showed the highest ex-vivo magnetometer counts. Furthermore, ferumoxytol had the highest lymph node retrieval rate. The effect of concentration, volume and time of harvesting was initially evaluated in the same porcine model by Ahmed et al. [25]. This study showed a significant positive correlation between magnetometer counts and iron content of excised SLNs (*r* = 0.82; *p* < 0.001) and increasing time of injection (*p* < 0.001), plateauing at 60 min. The total number of excised SLNs showed a significant positive correlation with increasing magnetic nanoparticle volumes (*p* < 0.001), and iron content in SLNs with increasing concentration (*p* = 0.006). In addition, Ahmed et al. [28] evaluated the impact of site and timing of magnetic nanoparticle injection in a murine model, showing a rapid uptake on MRI, smaller “void artifacts” (*p* < 0.001) and a significant increase in iron content with time in the group receiving subcutaneous injection (*r* = 0.94; *p* < 0.001). Previous clinical studies also evaluated different protocols of the SPIO injection. Hersi et al. [16], performed a study comparing patients from the Nordic SentiMag trial (2.0 mL Sienna+ injected on the morning of surgery) with two new patient groups: one group that received an injection of 1.0 mL Magtrace 1–7 days prior to surgery and a second group that received 1.5 mL Magtrace on the day of surgery. This study showed as well that a lower dose is also non-inferior to the standard technique with similar detection rates between the three doses (97.5% versus 100% and 97.6%, *p* = 0.11) and more SLNs excised than higher doses (2.18, versus 1.85 and 1.83, *p* = 0.003). Rubio et al. [19] compared the injection of 1.0, 1.5 and 2.0 mL Magtrace diluted with saline on the morning of surgery and found each group of 45 patients to be non-inferior to the standard technique (*p* = 0.654). Our study further evaluated the magnetic nanoparticle injection based on these experiments, focusing on evaluation of the effect of iron dose, dilution, time, and massaging the injection site. Time was an important factor, and longer waiting time ended up in more iron accumulation in the SLN within the timeframe investigated. With injection a day prior to surgery, people could reduce required amount of SPIO injection, which would reduce MRI artifact and coloration at the injection site. The results also indicated that iron accumulation in the SLN is proportional to the amount of iron injection between 20 ug (0.1 mg Iron/kg) and 200 ug (1 mg/kg) in this model. Although there must be interspecies differences, researchers may be able to explore injection of the current SPIO agents ranging between 0.2 and 2 mL with expectation of proportional accumulation in the SLN.

Recently, the first randomized trial [21] was published evaluating the magnetic technique for SLND on its own. The use of the magnetic tracer in one hospital was compared to the use of the radioisotope in a second hospital. A total of 338 patients (343 SLNDs) were included and identification rates of 95.6% for the magnetic technique and 96.9% for the radioisotope were found. The lymph node retrieval rate was 1.35 nodes per patient for the magnetic technique and 1.89 nodes per patient for the radioisotope technique. This study also looked at the timing of the injection and found that SPIO injections performed at a median of 16 days prior to SLND obtained a better identification rate (*p* = 0.031) and higher lymph node retrieval rate (*p* < 0.001) compared to injections on the morning of surgery. Other studies also suggested feasibility of preoperative injection up to weeks or months prior to surgery [22,23,29]. Although this study had been designed before these publications and examined preoperative injection up to 24 h, a longer waiting period might further reduce required injection volume of magnetic particles. This speculation should be investigated in future studies.

The current drawback of the magnetic technique using liquid tracers are the residual MRI artifacts seen in patients at the injection site post-operatively, even for years [30,31,32]. Within the SentiMAG and MagSNOLL trials [4,33] all patients who underwent an MRI after their SLND were asked to participate in a sub-study. The MRIs of these patients were evaluated, and it was seen that no artifact was visible in patients who participated in the MagSNOLL trial, but an artifact was visible in patients of the SentiMAG trial. This is most likely due to the resection of the injected area in the MagSNOLL trial. Our results indicated that the amount of residual iron at the injection site is proportional to injection dose, and accumulation of very small amount of iron can cause a void artifact on MRI. Although reduction in the injection dose would lead to a less severe artifact, it might be advisable that the injection site should be included in resection to avoid any ambiguity on future diagnostic imaging.

A major limitation of this study is that we used a rat model to optimize SPIO injection. Although it enabled us to address the factors associated with the injection, the SLNs of rats are small, and ex vivo counts by a magnetometer were not taken in this study. Therefore, the theoretical basis provided in this study should be further evaluated in more relevant animal models, which may include dog and pig, or in clinical studies. Preoperative injection more than one day, up to months, prior to surgery needs to be investigated in the future as well, to see if a longer waiting period facilitates more accumulation of iron in the SLNs and/or leads to complicating accumulation in secondary and tertiary nodes. In a recent study, patients were injected with SPIO prior to neoadjuvant chemotherapy, and MRI lymphography was compared before and after chemotherapy with a median of 130 days interval [29]. As a result, SPIO accumulation was observed in the same lymph nodes. Therefore, it was suggested that SPIO does not migrate in higher nodes for months, which would support SPIO injection more than 1 day prior to surgery. Also, subsequent studies can include visual images and MRI scanning of the injection site to address artifact and coloration around the area. Another limitation is that this study lacks sample size calculation. Again, the findings may need to be validated with the appropriate statistical power considering the pilot results from this study.

Tumor microenvironments evolve dynamically and continuously, shaping a niche in favor of tumor cell proliferation and dissemination [34,35]. A tissue structure is edited and distorted compared to its normal counterpart, and the lymphatic system is not an exception. Vigorous lymphangiogenesis and expansion of tumor cell nests are reported to lead to enlarged peritumor lymphatic vessels as well as collapsed intratumoral vessels [35]. As such, peritumorally injected SPIO would be put in different lymphatic dynamics compared to subcutaneous injection into the normal tissue performed in this study. Different lymphatic system, such as the axillary versus neck networks, may have different draining machineries. A more relevant tumor-bearing animal model and clinical studies will reveal these points in the future.

## 5. Conclusions

Iron accumulation in the SLNs was proportional to injection doses within a certain range. Time was also a primary factor for SLN uptake of SPIO. Dilution and massaging did not have significant effects. From these results, preoperative injection may be advantageous for SLN detection, as has been shown in the pioneering clinical studies, as well as for reduction in residual iron at the injection site by the expected decrease in required injection dose. The result from this study provides a theoretical background that helps our understanding of the magnetic SLND.

## Figures and Tables

**Figure 1 cancers-13-05031-f001:**
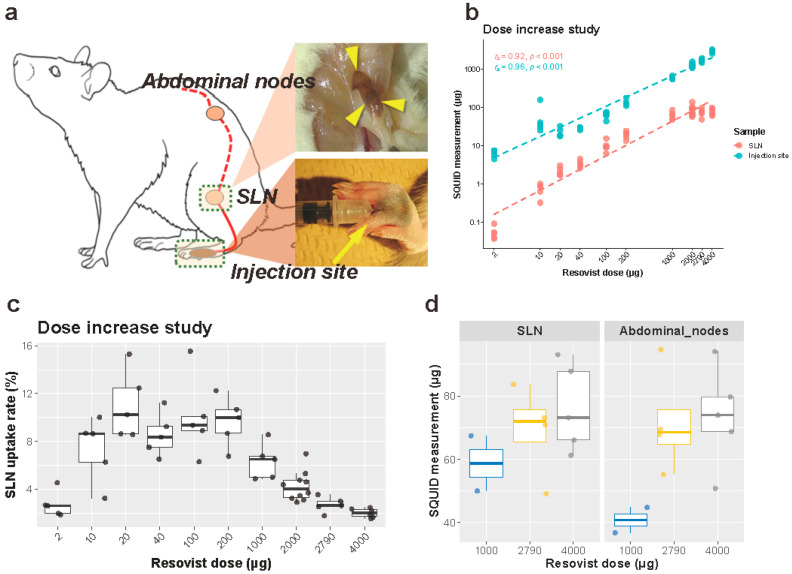
Schematic illustration of an animal study and iron dose increase experiment. (**a**) Schematic illustration of an animal study. (**b**) Iron accumulation measured by a superconducting quantum interference device (SQUID) in the SLNs (red) and at the injection sites (blue) 24 h after Resovist injection. Both axes are logarithmic scales. *n* = 5 each group, except for *n* = 10 of the 2000 µg group. (**c**) The SLN uptake rate in the dose increase experiment. Each measurement was overlaid on a box plot. (**d**) Iron accumulation in the secondary nodes in the 1000 µg (*n* = 2), 2790 µg (*n* = 4), and 4000 µg (*n* = 5) groups. Corresponding results from the SLN measurements are redisplayed on the left for comparison. SLN: sentinel lymph node.

**Figure 2 cancers-13-05031-f002:**
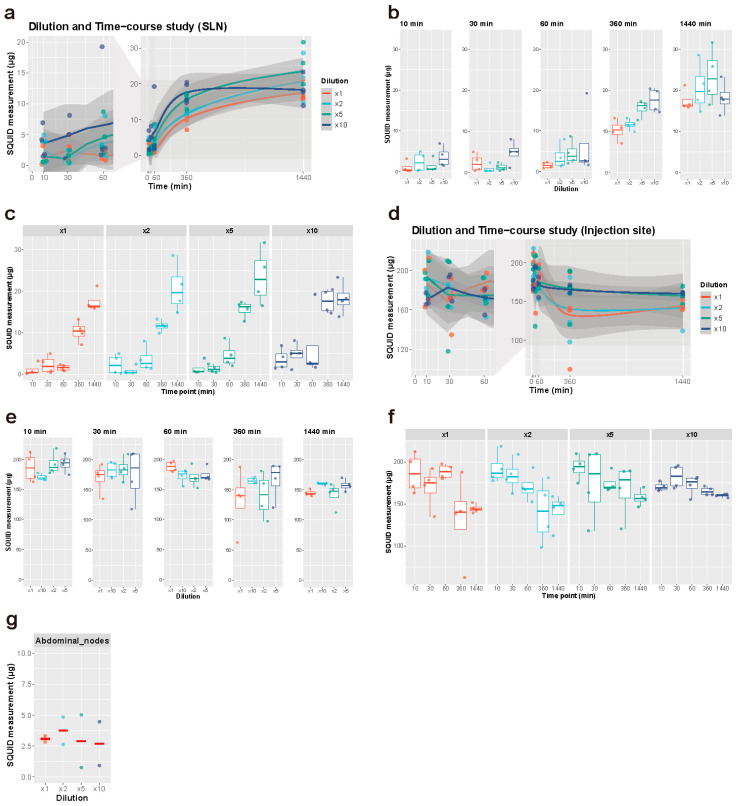
Dilution and time-course experiment. (**a**–**c**) Iron accumulation in the SLNs. (**d**–**f**) Iron accumulation at the injection sites. (**a**,**d**) All the measurements by SQUID were plotted along the time course. The solid lines represent regression curves in a LOESS model and the surrounding grey areas represent the confidence intervals. The time points between 0 and 60 min were magnified on the left. *n* = 4, each group. (**b**,**e**) Iron accumulation was compared between different dilutions at each time point. Each measurement (points) was overlaid on a box plot. (**c**,**f**) Iron accumulation was compared between different time points at each dilution. Each measurement (points) was overlaid on a box plot. (**g**) Iron accumulation in the secondary nodes 24 h after Resovist injection with varying dilution (*n* = 2). Corresponding results from SLN measurements can be found in **e** (1440 min). SLN: sentinel lymph node.

**Figure 3 cancers-13-05031-f003:**
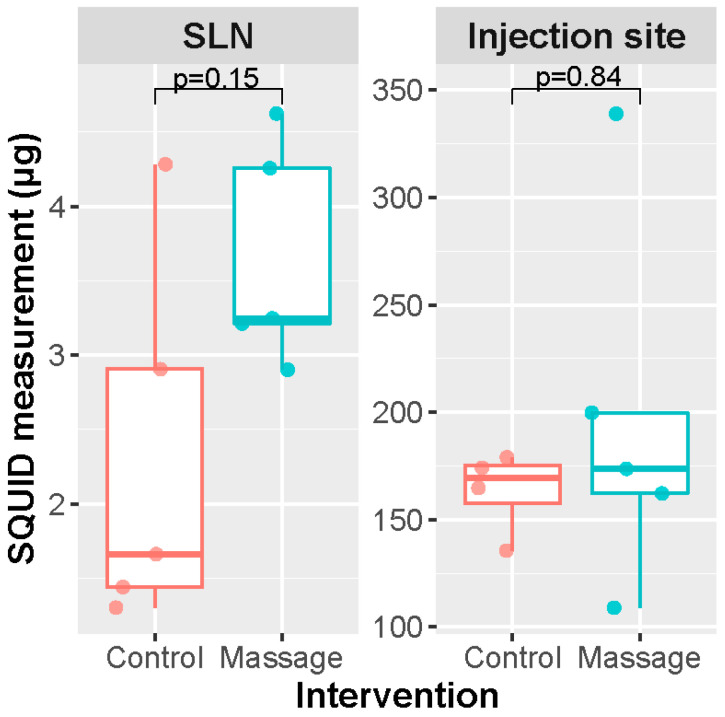
Results from massage experiment. *n* = 4, Mann–Whitney U test. SLN: sentinel lymph node.

**Figure 4 cancers-13-05031-f004:**
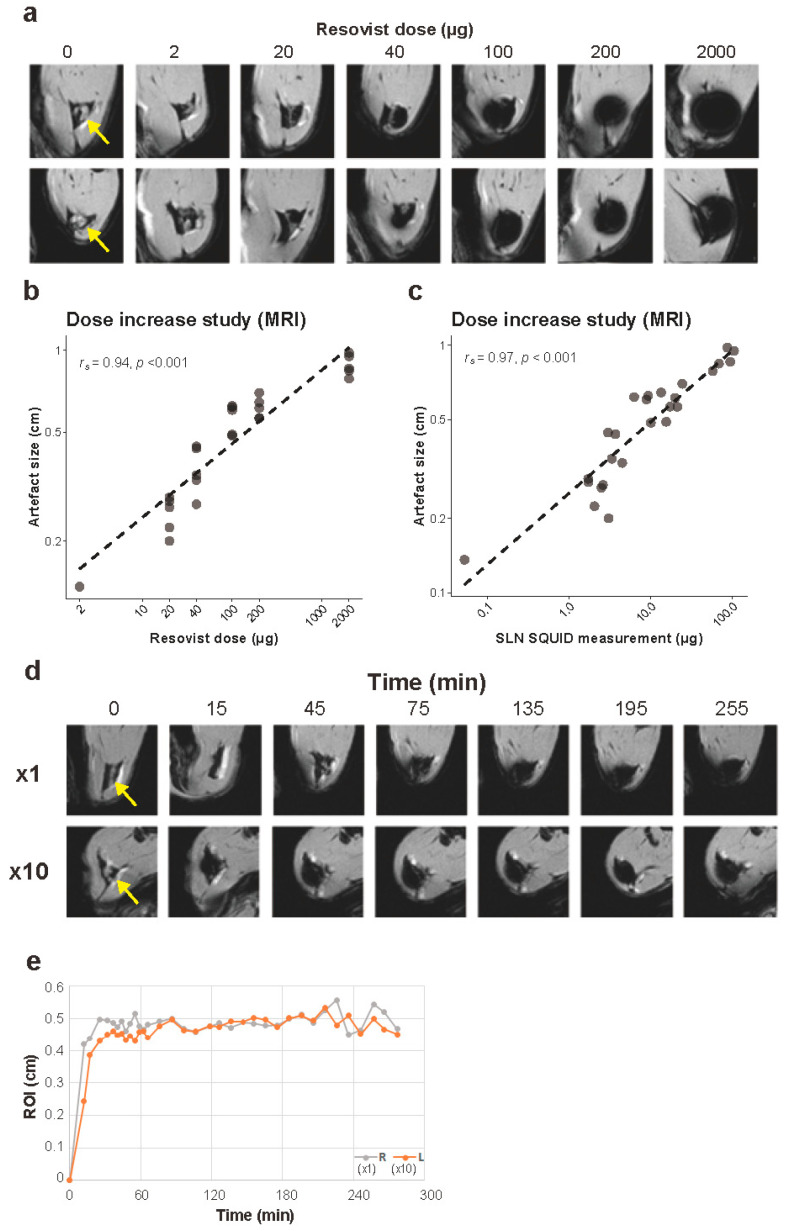
MRI experiment. (**a**) Representative transverse images of the rat hindlimbs at the popliteal lymph node level from magnetic resonance imaging (MRI). Yellow arrows indicate locations of the SLNs (popliteal lymph nodes) of control rats. (**b**) Relationship between the amount of Resovist injection and the size of artifacts measured on MRI. Both axes are on logarithmic scales, *n* = 5 for each group. (**c**) Relationship between iron accumulation measured by SQUID in SLNs and the size of artifact measured on MRI. Both axes are logarithmic scales. (**d**) Representative images from time-course MRI in a rat. (**e**) Time-course measurement of the size of artifact on MRI in a rat. MRI: magnetic resonance imaging.

## Data Availability

The data presented in this study are available on request from the corresponding author. The data are not publicly available due to ethical restrictions.

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
