# Peer review of "Optimization of SPIO Injection for Sentinel Lymph Node Dissection in a Rat Model"

_cancers, 2021, doi:10.3390/cancers13195031_

Round 1
Reviewer 1 Report
The manuscript by Peek et al explores the optimization of the magnetic tracer (SPIO) in a rat model. The manuscript has the potential to contribute, but needs major revisions before it can be considered for publication.
In detail:
- Introduction, p.2, r. 61-64 "However, no studies have had a comprehensive look at the optimal magnetic nanoparticles injection for SLNB, and questions remain regarding in body dynamics of the agent, to determine the optimum iron dose, optimum time course of injection, the efficacy of dilution and efficacy of massaging the injection site.": Regarding the time course of injection, there are studies that show that it has been tested to inject SPIO up to days to weeks before surgery. This is clearly stated in reference 24, that the authors use, and then there is the delayed SLND in DCIS (doi: 10.1002/bjs). Another study by the swedish group showed that preoperative injection increases the detection rate and the number of sentinels (doi: 10.1245/s10434-019-07239-5.) This means that the statement above needs to be modified and put into the context of this data. It becomes much more meaningful to discuss that, despite knowing already from clinical studies that preoperative injection is feasible, further optimization and refinement is beneficial.
- The authors adhere to the ARRIVE guidelines, but it would would be useful to refer to it and include a checklist. Additionally, state when the experiments were conducted.
- Is there any sample size calculation and, if not, why?
- It is unclear for the reader to follow how many animals did what and how many researchers did what to each animal group. A flowchart needs to be included to show the study flow clearly.
- The authors performed SLND (the procedure should be changed to SLND instead of SLNB throughout the text). They need to state how the SLNs were identified clearly (probe, procedure and so on).
- Statistical analysis p. 4, r. 181-190: Please clarify, because this is rather cloudy. Small numbers make non-parametric tests much more appropriate. If log conversion was used to fit in the normal distribution and then perform parametric tests and facilitate linear regression, then it should be named. If pre-log transformation outcomes are used, then non parametric comparisons should be the primary method. Descriptive statistics are needed as well as some type of control for normal distribution also. Please consult, cite and report according to the SAMPL guidelines (SAMPL Guidelines 3-13-13 (equator-network.org)).
- Please go through the results and check for consistency in definitions; it is really hard to follow.
- All the results on iron uptake in relation to dose and time of injection are very interesting, but what the manuscript is lacking and needs to be added is the impact of dose and time in the operative procedure. With other words, was SLN detection as easy with a lower dose? For this, some type of effect size is needed. If i.e. a probe was used for detection, were there differences in the counts? Such an analysis should be included. The authors themselves discuss it shortly (Discussion, p.11, r.383-384). I think that presenting results from the study data is very important.
- The "Results" section is difficult to follow; there is too much information overall which needs to be restructured. Additionally, there is too much information in the legends that should be in the text. A very typical example is p. 6-7 r. 267-277. Some of the most interesting results in the manuscript are given here and are almost "hidden" and unclear. Linear regression has been performed and the authors just give a p-value, stating that "the one variable explains the variance better than the other". However, just a p-value is inadequate. The authors must provide the coefficient with 95% confidence intervals, together with the p-value. It is very important to make this clear, as these results are highly relevant for clinical studies. They are very good results and need to "shine" in the manuscript.
- MRI artifacts: here, the authors again "hide" the important information that the Log of iron concentration is proportionate to the artifact size at the injection site but not to the artifact size in the SLN. This needs to be lifted.
- Discussion, p.11, r. 358-360: " However, further evaluation of the iron injection range between 200 and 1,000 μg is required. Utilization rates were low with injection of small iron volumes." What does this mean? Did it work in the animal model with these doses? What can one that performs a clinical trial expect and what would they need to take into account? This statement needs discussion.
- Discussion, p.11, r. 367-370: "During dilution experiments, longer time from injection increased iron uptake within popliteal nodes to a greater extent than increasing the diluting factor. This indicates that the lymphatic system may be able to take up more magnetic tracer if the injection is performed a day prior to surgery rather than injection just prior to surgery." Please refer to comment 1. There are already clinical data on the preoperative injection. I understand that experiments were performed with injections up to 24 hrs prior to SLND, but the technique is working with longer injection timepoints. So this needs to be discussed.
- Discussion, p.1 r. 377-378: " Massaging may facilitate iron uptake by the SLNs, but the effect was small and not statistically significant." This is in line with the results that the authors present. In the next paragraph, (p.12, r. 415-419), the "hypothetical recommendations" are contradictory. The authors state that " For patients who will have surgery on the same day as the injection,
massaging might be useful, and an iron volume diluted 5-10 times is recommended to optimize iron uptake. In patients who will have the injection on the day prior to surgery a dilution of 5-10 times is recommended, but massaging showed no additional benefit in terms of iron uptake in the popliteal nodes." The notion that massage may not have an effect in the iron concentration when everybody recommends vigorous massage for 5 minutes is very interesting and the authors are sending mixed signals here. Additionally, the "theoretical recommendation" part and any recommendation should be omitted. This is just a preclinical study and can help inform the design of a clinical trial. No recommendations can be made from a preclinical study on how to inject patients that will be operated in routine practice. Please remove this part and give a clear message on the findings about the massage. - Discussion, p.11, r. 379-382: " Although no human model was used for this study, it is possible to estimate the ideal iron volume, dilution and time of injection for SLNB in breast cancer patients by looking at the ratio between the previously evaluated and validated mouse and porcine models and the rat model evaluated in this study." How? This is very interesting. Please elaborate and provide a model.
- Discussion, p.12, r. 432-438. Regarding the artifacts, the authors cite what I find to be an abstract from a congress back in 2015. There are published reports and I think that they should be used instead or at least cited. These show the artifacts in the injection site which is under the areola in these cases. In the MagSNOLL patients received a smaller dose but it was injected in the lesion which was then removed. I think it is very clear that this is what explains the lack of artifacts and not the dose injected. Please revise accordingly. Then, the rest of the paragraph on magnetic markers has its value but is not clearly connected with what the authors are saying, so it should be rephrased.
- The conclusions are somewhat generic. Especially the last sentence about a randomized controlled trial, that is also present in the abstract, is not something that can be directly deduced by the study itself and should be omitted.
- Finally, one gets the feeling that the authors decided to experiment in the preclinical setting but chose not to test the time interval in more than 24hrs, despite that there is already clinical evidence that shows that preoperative injection is feasible and advantageous. It is as if the present results are out of phase. The authors should discuss why they did not consider that in their experiments, because, in its present form, it feels a little bit outdated.
- Small linguistic things such as artifacts/artefacts, anesthetized/anaesthetized. Please revise.
In conclusion, the manuscript has potential and, if all queries are adequately addressed, then it should be considered.
Author Response
Response to the comments from reviewer 1
- Introduction, p.2, r. 61-64 "However, no studies have had a comprehensive look at the optimal magnetic nanoparticles injection for SLNB, and questions remain regarding in body dynamics of the agent, to determine the optimum iron dose, optimum time course of injection, the efficacy of dilution and efficacy of massaging the injection site.": Regarding the time course of injection, there are studies that show that it has been tested to inject SPIO up to days to weeks before surgery. This is clearly stated in reference 24, that the authors use, and then there is the delayed SLND in DCIS (doi: 10.1002/bjs). Another study by the swedish group showed that preoperative injection increases the detection rate and the number of sentinels (doi: 10.1245/s10434-019-07239-5.) This means that the statement above needs to be modified and put into the context of this data. It becomes much more meaningful to discuss that, despite knowing already from clinical studies that preoperative injection is feasible, further optimization and refinement is beneficial.
Thank you for your great comment. We have revised the text in the introduction accordingly and included the study by Warnberg et al. and Karakatsanis et al. into the manuscript. As you might have imagined, this study had been planned before these publications around 2015. We agreed that longer waiting period should be tested in a similar experimental setting in the future. The following parts of the text have been revised:
Introduction, lines 62-64
Recently, studies have successfully employed injection days or even weeks prior to surgery, revealing that preoperative injection is also feasible [21–23].
Discussion, lines 434-438
Other studies also suggested feasibility of preoperative injection up to weeks prior to surgery [22,23]. Although this study had been designed before these publications and examined preoperative injection up to 24h, longer waiting period might further reduce required injection volume of magnetic particles. This speculation should be investigated in future studies.
Discussion, lines 456-459
Preoperative injection more than one day, up to months, prior to surgery needs to be in-vestigated in the future as well, to see if longer waiting period facilitates more accumula-tion of iron in the SLNs or leads to complicating accumulation in secondary and tertiary nodes.
- The authors adhere to the ARRIVE guidelines, but it would would be useful to refer to it and include a checklist. Additionally, state when the experiments were conducted.
Thank you for your instructions. The ARRIVE checklist has been completed and is included as supplementary data 1. In addition, the time of the experiments were added to the methods.
The following part of the text has been revised:
Materials and Methods lines 75-78.
Animal studies were approved by the local Ethics Board (Accession Number P15-124) and the experiments were performed between December 2015 and March 2016 at the University of Tokyo according to the guidelines of the institution and ensured humane care of animals. Authors adhered to the ARRIVE guidelines (Data S1).
- Is there any sample size calculation and, if not, why?
Thank you for pointing it out. We failed to explicitly state this. The number of animals in each group (n=4~5) was determined according to the previous publication where optimization was attempted in a porcine model (Ahmed, M.; Anninga, B.; Pouw, J.J.; Vreemann, S.; Peek, M.; Van Hemelrijck, M.; Pinder, S.; Ten Haken, B.; Pankhurst, Q.; Douek, M. Optimising Magnetic Sentinel Lymph Node Biopsy in an in Vivo Porcine Model. Nanomedicine 2015, 11, 993–1002, doi:10.1016/j.nano.2015.01.010.). No sample size calculation was performed as there was no preceding study using a rat model and Resovist. We put this point in the Discussion (limitation) as well.
The following part of the text has been revised:
Materials and Methods lines 78-80.
No sample size calculation was performed and the number of animals in each group (n=4~5) was determined according to the previous publication [24].
Discussion lines 461-463
Another limitation is that this study lacks sample size calculation. Again, the findings may need to be validated with the appropriate statistical power considering the pilot results from this study.
- It is unclear for the reader to follow how many animals did what and how many researchers did what to each animal group. A flowchart needs to be included to show the study flow clearly.
Thank you for your suggestion. A flow diagram has been added as supplementary material, figure S1. 55 animals were used for the dose increase experiments, of which 30 were subsequently used for the MR experiments. 2 animals were used as controls for these MR experiments. 45 animals were used for the time and dilution experiments. 5 rats were used for the massage experiments and 1 animal was used for the MR time experiments.
The following part of the text has been revised:
Materials and Methods lines 82.
A flow diagram of the entire study can be found in Figure S1.
- The authors performed SLND (the procedure should be changed to SLND instead of SLNB throughout the text). They need to state how the SLNs were identified clearly (probe, procedure and so on).
Thank you for your great comment. In this model, the popliteal lymph node was visually identified and dissected at its anatomical location. Signals from rat’s lymph node is too weak to be detected by a magnetometer used for human medicine. The following part of the text has been revised:
Materials and Methods lines 108-109.
The caudal skin of the stifle joint was incised, and the popliteal lymph nodes were visually identified at its anatomical location (Figure 1a).
Regarding terminology, the authors believe that SLNB is also a general term used for removal of the SLN’s in in vivo animal studies without tumor implantation. To our knowledge, the majority of the articles on this subject use SLNB [18, 21, 22, 23]. Therefore, we would like to keep “SLNB” for consistency. Instead, we introduced the term “sentinel lymph node dissection” with SLNB when it appears first. The following part of the text has been revised:
Introduction lines 39-41.
Sentinel lymph node biopsy (SLNB), also known as sentinel lymph node dissection, is the standard of care for clinically and radiologically node-negative breast cancer pa-tients to stage the axilla and determine if cancer has spread to regional lymph nodes [1,2].
- Statistical analysis p. 4, r. 181-190: Please clarify, because this is rather cloudy. Small numbers make non-parametric tests much more appropriate. If log conversion was used to fit in the normal distribution and then perform parametric tests and facilitate linear regression, then it should be named. If pre-log transformation outcomes are used, then non parametric comparisons should be the primary method. Descriptive statistics are needed as well as some type of control for normal distribution also. Please consult, cite and report according to the SAMPL guidelines (SAMPL Guidelines 3-13-13 (equator-network.org)).
Thank you very much for bringing up this. We agreed that nonparametric tests are more appropriate. We removed descriptions of parametric tests (Pearson correlation and t-test) and their results. Regarding multiple regression analysis, we added the detailed description in the Result section. During this revision, we noticed that we had mistakenly considered dilution as a statistically significant factor for iron amount at the injection site in the previous version. Related sentences were revised as well. We also refer to the SAMPL guideline and added Supplementary tables 1 and 2 for descriptive statistics. The following part of the text have been revised:
Materials and Methods lines 189-200.
2.6. Statistical analysis
Nonparametric (2-sided) tests were used wherever possible. The correlation between iron accumulation in the SLNs and the injection sites, and the injected dose was calculatd using the Spearman (rank-based) correlation. Statistical differences in iron uptake be-ween massage and no-massage groups were assessed using the Mann-Whitney U-test. The Spearman correlation was also used to examine associations between MRI artifact size and iron uptake, dilution or time of injection. A parametric analysis had to be used to simultaneously regress iron accumulation in the SLNs on dilution, time, and the amount of iron at the injection site. This multiple regression is described later.
All statistical analysis and visualization were performed using IBM SPSS Statistics Version 23 (SPSS Inc., USA) and results were reported according to the SAMPL guideline[25].
Results lines 274-295.
To find the best single predictors of iron volume in the SLN and the injection site, Spearman correlations were calculated. Correlation coefficients of iron volume in the SLN with dilution, time, and iron volume at the injection sites are 0.201 (P = 0.07), 0.810 (P < 0.001), and -0.624 (P < 0.001), respectively. This result showed that time is a strong predictor for iron volume in the SLN with statistical significance. To study the uptake of iron at the SLN, it was necessary to regress SLN iron accumulation on time, dilution, and iron concentration at the injection site. For the purpose, linear and squared terms of the predictors were used. In addition, the SLN iron concentration needed to be transformed. The Box-Cox transformation y = (xα-1)/α was used, which reduces to a log transformation as α → 0. For the SLN regression, α had a 95% confidence interval of (0.3, 0.59). Hence, for simplicity, the square root of SLN iron volume was used in the regression. Detailed results are given in Table S3, and confirmed that dilution was not a significant predictor. Time was, and also time squared, with a negative coefficient, showing that iron accumulation increases sublinearly, i.e. it slows down with time. Turning to iron volume at the injection site, the correlation coefficients with dilution and time are 0.0108 (P = 0.92) and -0.554 (P < 0.001), respectively. Again, time is the best predictor, and dilution has no significant effect. For multiple regression, the iron volume at the injection site was fitted with α = 1. As a result, dilution was not significant, whereas again the decrease with time was significant, and also slowing down significantly with time. Full results are given in Table S4. Consequently, time after injection is the most significant factor for both increasing iron accumulation in the SLN and facilitating iron clearance at the injection site. Dilution does not have any significant effects on both.
- Please go through the results and check for consistency in definitions; it is really hard to follow.
Thank you for your comment. The authors have gone through the entire manuscript to revise inconsistent phrases/words for improved readability. For example, we found multiple words were used to describe one thing, which may have caused impaired readability. The representative revisions were as follows:
Popliteal lymph nodes, SLN -> “SLN”
Abdominal lymph nodes, secondary nodes -> “Secondary nodes”
Injection site, leg -> “Injection site”
Iron volume, iron dose -> “iron dose”
Other than these, the authors carefully revised the entire manuscript. I hope that this revision made the manuscript easy to follow.
- All the results on iron uptake in relation to dose and time of injection are very interesting, but what the manuscript is lacking and needs to be added is the impact of dose and time in the operative procedure. With other words, was SLN detection as easy with a lower dose? For this, some type of effect size is needed. If i.e. a probe was used for detection, were there differences in the counts? Such an analysis should be included. The authors themselves discuss it shortly (Discussion, p.11, r.383-384). I think that presenting results from the study data is very important.
Thank you for your valuable comment. The aim of this study is to optimize SPIO injection and know which factors affect iron volume in the SLN most. We have chosen rats because we tried to address the issue rather comprehensively. However, on the other hand, the lymph nodes of the rats are not big to accommodate enough iron to be detected by a magnetometer used in human medicine. Therefore, this part needs further studies using more relevant animal models (i.e. dog, pig, etc) in which a magnetometer can be used or clinical studies. We addressed this point in the Discussion as a limitation of the study.
Discussion lines 451-456.
A major limitation of this study is that we used a rat model to optimize SPIO injection. Although it enabled us to comprehensively address the factors associated with the injection, the lymph nodes cannot accommodate large amount of iron to be detected by a magnetometer currently used in human medicine. Therefore, the theoretical basis provided in this study should be further evaluated in more relevant animal models, which may include dog and pig, or in clinical studies.
- The "Results" section is difficult to follow; there is too much information overall which needs to be restructured. Additionally, there is too much information in the legends that should be in the text. A very typical example is p. 6-7 r. 267-277. Some of the most interesting results in the manuscript are given here and are almost "hidden" and unclear. Linear regression has been performed and the authors just give a p-value, stating that "the one variable explains the variance better than the other". However, just a p-value is inadequate. The authors must provide the coefficient with 95% confidence intervals, together with the p-value. It is very important to make this clear, as these results are highly relevant for clinical studies. They are very good results and need to "shine" in the manuscript.
Thank you very much for your great advice. The results section has been divided into sub-sections for easier interpretation. The legends of the figures have been shortened. The entire manuscript has been revised for improved readability as explained in our response to your comment 7.
Regarding the results of the linear regression, we agreed with your assessment. We provided the detailed results in Tables S3 and S4. We are pleased that you found the results clinically relevant. We revised the text to clarify our points as follows.
Results, lines 273-295
3.2.3 Statistical analysis
To find the best single predictors of iron volume in the SLN and the injection site, Spearman correlations were calculated. Correlation coefficients of iron volume in the SLN with dilution, time, and iron volume at the injection sites are 0.201 (P = 0.07), 0.810 (P < 0.001), and -0.624 (P < 0.001), respectively. This result showed that time is a strong predictor for iron volume in the SLN with statistical significance. To study the uptake of iron at the SLN, it was necessary to regress SLN iron accumulation on time, dilution, and iron concentration at the injection site. For the purpose, linear and squared terms of the predictors were used. In addition, the SLN iron concentration needed to be transformed. The Box-Cox transformation y = (xα-1)/α was used, which reduces to a log transformation as α → 0. For the SLN regression, α had a 95% confidence interval of (0.3, 0.59). Hence, for simplicity, the square root of SLN iron volume was used in the regression. Detailed results are given in Table S3, and confirmed that dilution was not a significant predictor. Time was, and also time squared, with a negative coefficient, showing that iron accumulation increases sublinearly, i.e. it slows down with time. Turning to iron volume at the injection site, the correlation coefficients with dilution and time are 0.0108 (P = 0.92) and -0.554 (P < 0.001), respectively. Again, time is the best predictor, and dilution has no significant effect. For multiple regression, the iron volume at the injection site was fitted with α = 1. As a result, dilution was not significant, whereas again the decrease with time was significant, and also slowing down significantly with time. Full results are given in Table S4. Consequently, time after injection is the most significant factor for both increasing iron accumulation in the SLN and facilitating iron clearance at the injection site. Dilution does not have any significant effects on both.
- MRI artifacts: here, the authors again "hide" the important information that the Log of iron concentration is proportionate to the artifact size at the injection site but not to the artifact size in the SLN. This needs to be lifted.
Thank you very much for pointing this out. We apology that our descriptions were hard to follow. In this study, the aim was to optimize accumulation in the SLN. Thus, we scanned SLNs, but not injected sites. We were not able to scan both with time and instrumental restrictions. So, we correlated the artifact at the SLNs with injection dose and absolute amount of iron in the resected SLNs. We modified the text to make this clear. We understand that artifact at the injection site is as important as that of SLNs. We put this in the Discussion as a limitation.
Results, lines 325-333.
The SLNs of 30 rats from the dose increase experiment and two controls were successfully scanned with MRI immediately after euthanasia. On T1-weighed images, SPIO accumula-tion in the SLNs was observed as a signal reduction of the entire lymph node in the 2-μg group and as a spherical artifact in the other five groups (Figure 4a). Control groups showed no signal reduction on MRI.
The size of the artifact at the SLNs increased with the injected volume of iron (Spear-man’s rs(30)=0.94, p<0.01, Figure 4b). In addition, a statistically significant correlation was found between the amount of iron in the SLNs determined by the SQUID and the artifact size (Spearman’s rs(30)=0.97, p<0.01, Figure 4c).
Discussion, lines 459-461.
Subsequent studies can include visual images and MRI scanning of the injection site to address artifact and coloration around the area.
- Discussion, p.11, r. 358-360: " However, further evaluation of the iron injection range between 200 and 1,000 μg is required. Utilization rates were low with injection of small iron volumes." What does this mean? Did it work in the animal model with these doses? What can one that performs a clinical trial expect and what would they need to take into account? This statement needs discussion.
Thank you very much for this important comment. As we reviewed the manuscript, we found that this part does not mean what we wanted to convey. We tried to say that there is a certain range where people can expect proportional uptake by the SLN to the injection dose. Therefore, people do not want to try too much or too low injection of SPIO in clinical trials. Accordingly, we revised the text as follows:
Discussion, lines 362-369.
This study, by investigating wide range of SPIO injection volume, revealed both a gradual increase of iron in SLNs along with an increase in iron injection dose and an existence of a plateau uptake for the first time in one experimental setting. Proportional increase in iron in the SLNs was observed between 20 (0.1 mg Iron/BW in a 200 g rat) and 200 μg (1 mg Iron/kg BW) iron injection. For a person who weighs 60 kg, 2 mL of neat SPIO (either of Sienna+, Magtrace, or Resovist) injection equals to 1 mg Iron/kg BW, approximately. Utilization rates were also low with injection of small iron volumes less than 10 μg.
- Discussion, p.11, r. 367-370: "During dilution experiments, longer time from injection increased iron uptake within popliteal nodes to a greater extent than increasing the diluting factor. This indicates that the lymphatic system may be able to take up more magnetic tracer if the injection is performed a day prior to surgery rather than injection just prior to surgery." Please refer to comment 1. There are already clinical data on the preoperative injection. I understand that experiments were performed with injections up to 24 hrs prior to SLND, but the technique is working with longer injection timepoints. So this needs to be discussed.
Thank you very much and we all agreed with your points. Together with our response to your comment 1, the following part of the text have been revised:
Discussion, lines 377-382.
During dilution experiments, longer time from injection increased iron uptake within popliteal nodes to a greater extent than increasing the dilution factor. This indicates that the lymphatic system may be able to take up more magnetic tracer if the injection is per-formed a day prior to surgery rather than injection just prior to surgery, as has already been shown in some studies [16,21–23]. For iron at the injection site, time is also a signifi-cant factor for washout.
Discussion, lines 434-438.
Other studies also suggested feasibility of preoperative injection up to weeks prior to surgery [22,23]. Although this study had been designed before these publications and examined preoperative injection up to 24h, longer waiting period might further reduce required injection volume of magnetic particles. This speculation should be investigated in future studies.
Discussion, lines 456-459.
Preoperative injection more than one day, up to months, prior to surgery needs to be investigated in the future as well, to see if longer waiting period facilitates more accumulation of iron in the SLNs or leads to complicating accumulation in secondary and tertiary nodes.
- Discussion, p.1 r. 377-378: " Massaging may facilitate iron uptake by the SLNs, but the effect was small and not statistically significant." This is in line with the results that the authors present. In the next paragraph, (p.12, r. 415-419), the "hypothetical recommendations" are contradictory. The authors state that " For patients who will have surgery on the same day as the injection, massaging might be useful, and an iron volume diluted 5-10 times is recommended to optimize iron uptake. In patients who will have the injection on the day prior to surgery a dilution of 5-10 times is recommended, but massaging showed no additional benefit in terms of iron uptake in the popliteal nodes." The notion that massage may not have an effect in the iron concentration when everybody recommends vigorous massage for 5 minutes is very interesting and the authors are sending mixed signals here. Additionally, the "theoretical recommendation" part and any recommendation should be omitted. This is just a preclinical study and can help inform the design of a clinical trial. No recommendations can be made from a preclinical study on how to inject patients that will be operated in routine practice. Please remove this part and give a clear message on the findings about the massage.
Thank you very much for your comment. We agree that we should not state any clinical recommendations. Instead, we replaced them with simple messages drawn by the results.
Discussion, lines 414-423.
Our study, further evaluated the magnetic nanoparticle injection based on these experiments, focusing on evaluation of the effect of iron dose, dilution, time, and massaging the injection site. Time was an important factor, and longer waiting time ended up in more iron accumulation in the SLN within the timeframe investigated. With injection a day prior to surgery, people could reduce required amount of SPIO injection, which would reduce MRI artifact and coloration at the injection site. The results also indicated that iron accumulation in the SLN is proportional to the amount of iron injection between 20 ug (0.1 mg Iron/kg) and 200 ug (1 mg/kg) in this model. Although there must be interspecies differences, people may be able to explore injection of the current SPIO agents ranging 0.2 and 2 mL with expectation of proportional accumulation in the SLN. - Discussion, p.11, r. 379-382: " Although no human model was used for this study, it is possible to estimate the ideal iron volume, dilution and time of injection for SLNB in breast cancer patients by looking at the ratio between the previously evaluated and validated mouse and porcine models and the rat model evaluated in this study." How? This is very interesting. Please elaborate and provide a model.
Thank you for pointing out this part. We realized that this is clearly overstated. We never be able to estimate a human model from the current results. We revised this part as follows:
Discussion, lines 385-387.
Although no human model was used for this study, it may be possible to suggest critical factors and conditions to optimize SPIO injection for breast cancer patients, together with the previous studies.
(At the end of the same paragraph, this statement is followed up by the sentences below as explained in our response to your previous comment.)
Discussion, lines 416-423.
Time was an important factor, and longer waiting time ended up in more iron accumulation in the SLN within the timeframe investigated. With injection a day prior to surgery, people could reduce required amount of SPIO injection, which would reduce MRI artifact and coloration at the injection site. The results also indicated that iron accumulation in the SLN is proportional to the amount of iron injection between 20 ug (0.1 mg Iron/kg) and 200 ug (1 mg/kg) in rats. Although there must be interspecies differences, people may be able to explore injection of the current SPIO agents ranging 0.2 and 2 mL with expectation of proportional accumulation in the SLN.
- Discussion, p.12, r. 432-438. Regarding the artifacts, the authors cite what I find to be an abstract from a congress back in 2015. There are published reports and I think that they should be used instead or at least cited. These show the artifacts in the injection site which is under the areola in these cases. In the MagSNOLL patients received a smaller dose but it was injected in the lesion which was then removed. I think it is very clear that this is what explains the lack of artifacts and not the dose injected. Please revise accordingly. Then, the rest of the paragraph on magnetic markers has its value but is not clearly connected with what the authors are saying, so it should be rephrased.
Thank you very much for your valuable comments. We all agreed with your points. Accordingly, we included three new published references.
Arslan, G.; Yılmaz, C.; Çelik, L.; Çubuk, R.; Tasalı, N. Unexpected Finding on Mammography and MRI Due to Accumulation of Iron Oxide Particles Used for Sentinel Lymph Node Detection. Eur J Breast Health 2019, 15, 200–202, doi:10.5152/ejbh.2019.4410.
30. Aribal, E.; Çelik, L.; Yilmaz, C.; Demirkiran, C.; Guner, D.C. Effects of Iron Oxide Particles on MRI and Mammography in Breast Cancer Patients after a Sentinel Lymph Node Biopsy with Paramagnetic Tracers. Clin Imaging 2021, 75, 22–26, doi:10.1016/j.clinimag.2020.12.011.
31. Krischer, B.; Forte, S.; Niemann, T.; Kubik-Huch, R.A.; Leo, C. Feasibility of Breast MRI after Sentinel Procedure for Breast Cancer with Superparamagnetic Tracers. European Journal of Surgical Oncology 2018, 44, 74–79, doi:10.1016/j.ejso.2017.11.016.
We also revised the text based on your suggestion. Regarding magnetic markers for localization, we realized that it is not relevant in this context, as you thought. These sentences were removed. Conclusively, the text has been revised as follows:
Discussion, lines 439-450.
The current drawback of the magnetic technique using liquid tracers are the residual MRI artifacts seen in patients at the injection site post-operatively, even for years [29–31]. Within the SentiMAG and MagSNOLL trials [4,32] all patients who underwent an MRI after their SLNB were asked to participate in a sub-study. The MRIs of these patients were evaluated, and it was seen that no artifact was visible in patients who participated in the MagSNOLL trial, but an artifact was visible in patients of the SentiMAG trial. This is most likely due to the resection of the injected area in the MagSNOLL trial. Our results indicated that the amount of residual iron at injection site is proportional to injection dose, and accumulation of very small amount of iron can cause void artifact on MRI. Although reduction of injection dose would lead to less severe artifact, it might be advisable that the injection site should be included in resection to avoid any ambiguity on future diagnostic imaging.
- The conclusions are somewhat generic. Especially the last sentence about a randomized controlled trial, that is also present in the abstract, is not something that can be directly deduced by the study itself and should be omitted.
Thank you very much for your instruction. We agreed with your opinions and revised the conclusions, as well as Simple Summary and Abstract, accordingly. The following part of the text have been revised:
Simple Summary, lines 18-21.
In this study, following injection characteristics were evaluated to optimize magnetic tracer uptake in the sentinel lymph nodes (SLN) in a rat hindleg model: (a) iron dose volume, (b) effect of dilution, (c) effect of injecting at different time courses and (d) effect of massaging the distal hindlegs. In conclusion, injection dose and time were primary factors for the SLN iron uptake.
Abstract, lines 22-34.
The magnetic technique, consisting of a magnetic tracer and a handheld magnetometer, is a promising alternative technique for sentinel lymph node biopsy (SLNB) and was shown to be non-inferior to the standard technique in terms of identification rates. In this study, injection characteristics (iron dose, dilution, time course and massaging) were evaluated to optimize magnetic tracer uptake in the sentinel lymph nodes (SLN) in a rat hindleg model. 202 successful SLNBs were performed. Iron uptake in the SLN is proportional (10% utilization rate) to the injection dose between 20- and 200 μg, showing a plateau uptake of 80 μg in the SLN around 1,000 μg injection. Linear regression showed that time had a higher impact than dilution, on the SLN iron uptake. Massaging showed no significant change on iron uptake. The amount of residual iron at the injection site was also proportional to the injection dose without any plateau. Time was a signif-icant factor for wash out of residual iron. From these results, preoperative injection may be ad-vantageous for SLN detection as well as reduction in residual iron at the injection site by potential decrease in required injection volume.
Conclusions, lines 474-479.
Iron accumulation in the SLNs were proportional to injection doses in the certain range. Time is also a primary factor for SLN uptake of SPIO. Dilution and massaging did not have significant effects. From these results, preoperative injection may be advantageous for SLN detection as well as reduction in residual iron at the injection site by potential decrease in required injection volume. Probable benefits from longer waiting period more than one day should be tested in the future. - Finally, one gets the feeling that the authors decided to experiment in the preclinical setting but chose not to test the time interval in more than 24hrs, despite that there is already clinical evidence that shows that preoperative injection is feasible and advantageous. It is as if the present results are out of phase. The authors should discuss why they did not consider that in their experiments, because, in its present form, it feels a little bit outdated.
We appreciate your comments so much. They have definitely improved scientific accuracy and readability of the manuscript. In association with your comment 1 and 12, this experiment was designed before these publications and it is major limitation of the study. We should have stated this clearly. We revised the text accordingly, as follows:
Discussion, lines 434-438.
Other studies also suggested feasibility of preoperative injection up to weeks prior to surgery [22,23]. Although this study had been designed before these publications and examined preoperative injection up to 24h, longer waiting period might further reduce required injection volume of magnetic particles. This speculation should be investigated in future studies.
Discussion, lines 456-459.
Preoperative injection more than one day, up to months, prior to surgery needs to be investigated in the future as well, to see if longer waiting period facilitates more accumulation of iron in the SLNs or leads to complicating accumulation in secondary and tertiary nodes. - Small linguistic things such as artifacts/artefacts, anesthetized/anaesthetized. Please revise.
Thank you very much for your careful reviewing. These have been revised to the American English version of both words.

Reviewer 2 Report
This manuscript is about optimization of SPIO injection as part of a magnetic sentinel node biopsy procedure. The authors used a rat model and evaluated several variables which may have impact on SPIO dose in sentinel lymph nodes and MRI artefacts at the injections site. Studies are well performed and described.
I have only some general comments.
Peritumoral injections may be different from subcutaneous injections in normal tissue. The micro-environment around the tumor is different from normal tissue, with higher amount of cells and different cells. Please comment in discussion on potential differences for the results found in this study.
The sentinel lymph nodes were they harvested directly guided by the magnetometer, or harvested and checked by the magnotometer?
In discussion is mentioned that the maximum accomodation depend on size of lymph node. It is suggested that tis means animal versus human. But also in human lymph node size may differ from nodal basin to nodal basin. Please add comment.
In discussion in it is mentioned that there are differences in lympahtic systems from animals and humans, but also in humans the lymphatic system may be different for axilla and neck for example. Please add comment.
Author Response
Responses to reviewer 2
This manuscript is about optimization of SPIO injection as part of a magnetic sentinel node biopsy procedure. The authors used a rat model and evaluated several variables which may have impact on SPIO dose in sentinel lymph nodes and MRI artefacts at the injections site. Studies are well performed and described.
Thank you very much for your careful reviewing. We believe that the comments from both reviewers improved the manuscript a lot. Please be notified that, after revising the statistical part of the manuscript, we noticed that dilution is no longer a significant factor for iron at the injection site. Therefore, we revised the related parts. We hope this revision answer to your concerns appropriately.
I have only some general comments.
- Peritumoral injections may be different from subcutaneous injections in normal tissue. The micro-environment around the tumor is different from normal tissue, with higher amount of cells and different cells. Please comment in discussion on potential differences for the results found in this study.
Thank you very much for this important comment. We completely lacked this perspective in the previous version. According to your comment, we revised a part of discussion as follows.
Discussion, lines 463-472.
Tumor microenvironment evolves dynamically and continuously, shaping a niche in favor of tumor cell proliferation and dissemination[32,33]. A tissue structure is edited and distorted compared to its normal counterpart, and lymphatic system is not an exception. Vigorous lymphangiogenesis and expansion of tumor cell nests are reported to lead to enlarged peritumor lymphatic vessels as well as collapsed intratumoral vessels[33]. As such, peritumoral injection of SPIO would have different consequences from subcutaneous injection into the normal tissue performed in this study. Different lymphatic system, such as the axillary versus neck networks, may have different draining machineries. A more relevant tumor-bearing animal model and clinical studies will reveal these points in the future.
- The sentinel lymph nodes were they harvested directly guided by the magnetometer, or harvested and checked by the magnotometer?
Thank you for pointing this out. In this model, the popliteal lymph node was visually identified and dissected at its anatomical location. Signals from rat’s lymph node is too weak to be detected by a magnetometer used for human medicine. The following part of the text has been revised:
Materials and Methods lines 108-109.
The caudal skin of the stifle joint was incised, and the popliteal lymph nodes were visually identified at its anatomical location (Figure 1a).
- In discussion is mentioned that the maximum accomodation depend on size of lymph node. It is suggested that tis means animal versus human. But also in human lymph node size may differ from nodal basin to nodal basin. Please add comment.
The discussion has been amended accordingly, including this feedback.
Thank you for this valuable comment. We revised the text accordingly, as follows:
Discussion, lines 356-358.
The maximum accommodation of iron in lymph nodes may depend on size of lymph node. It may differ not only between different animal species, but between individuals of the same species and the locations of the nodes in the body. Size of SPIO particles may matter as well.
- In discussion in it is mentioned that there are differences in lympahtic systems from animals and humans, but also in humans the lymphatic system may be different for axilla and neck for example. Please add comment.
Thank you again for another great comment. We revised the text accordingly, as follows:
Discussion, lines 470-472.
Different lymphatic system, such as the axillary versus neck networks, may have different draining machineries. A more relevant tumor-bearing animal model and clinical studies will reveal these points in the future.
Round 2
Reviewer 1 Report
The authors have addressed the majority of my comments in a satifactory manner and have provided a greatly improved manuscript. Some further points must be revised, however, so as to provide a consistent and relevant manuscript.
These are the remaining points to address.
1) p2, r64-69: "However, none of these studies have had a comprehensive look at the optimal magnetic nanoparticles injection for SLNB, and questions remain regarding in body dynamics of the agent, to determine the optimum iron dose, optimum time course of injection, the efficacy of dilution and efficacy of massaging the injection site. These factors are important in order to reduce the residual magnetic nanoparticles left at the injection site. "
The "comprehensive look" is a little bit unclear of a formulation. These are clinical studies and they had a different endpoint; it would probably not be possible to have ethical approval to study the body dynamics of SPIO on humans and this is why this paper has its role. Please rephrase, as it now seems that you are "questioning" the studies you are citing, whereas it is not so.
2. p.3, r 108-109: "The caudal skin of the stifle joint was incised, and the popliteal lymph nodes were visually identified at its anatomical location".
In your response, you write that no probe was used to identify the nodes, if I understand correctly. This means that "visually identified" refers to color? Please clarify. Additionally, if you did not take ex vivo counts, please clarify as it is a limitation, given the fact that, for clinicians that would use a magnetic biopsy, it would be harder to correlate to amount of iron; instead they would think of probe counts and color, such as in the case of isotope and blue dye.
3. p5, r 229-232. "A preliminary observation indicated that iron accumulation in the secondary nodes would increase with the iron injection dose as in SLNs (Figure 1d), although statistical testing was not performed due to the limited number of animals involved."
Since no statistical testing was performed, this needs to be removed completely.
4. p12, r456-458: "Preoperative injection more than one day, up to months, prior to surgery needs to be investigated in the future as well, to see if longer waiting period facilitates more accumulation of iron in the SLNs and/or leads to complicating accumulation in secondary and tertiary nodes. "
This is very relevant. Looking into the literature, there is this recent publication (Jazrawi et al, Cancers 2021, 13(17), 4285; https://doi.org/10.3390/cancers13174285). Here, some patients were injected with SPIO prior to neoadjuvant therapy and were explored with axillary MRI before and after. Then, surgery was performed after neoadjuvant with the coadministration of isotope. The authors found that the same nodes were visualized and detected by both methods. This makes the case that it probably does not migrate higher. Please discuss this.
5. p.11, r.403-404: "Some articles looked at aspects of the SPIO injection, but no comprehensive clinical studies were performed."
Again, "comprehensive" is rather vague and the comment is inaccurate. The studies by Rubio and Hersi that you cite look into the effect of dose, injection site and injection site. So they are as comprensive as it can be, given that they are clinical trials. Please omit.
6. p.13, r 468-471: "As such, peritumoral injection of SPIO would have different consequences from subcutaneous injection into the normal tissue performed in this study. Different lymphatic system, such as the axillary versus neck networks, may have different draining machineries."
How would it have consequences? Do we know it? Or is it an assumption? Please revise.
7. p 13, r.478-479: "Probable benefits from longer waiting period more than one day should be tested in the future."
You are already citing studies with demonstrated benefits from a much longer waiting period. So this does not need to be tested. Revise in a manner that contains this message. The experiments were conducted previously, but they are presented now, so the results need to placed in this context accurately. As with comment#5, the point is that the writing style in some points looks somewhat aggressive towards the cited literature, as if trying to deconstruct it, in order to show what has not been done yet and what needs to be done. It is my understanding that this has previously been a more popular writing style, but it is very important to be able to set things into scientific context. This means that claims need to be completely transparent, justified and evidence-based. I cannot stress how important this is for the value and the appropriateness of any manuscript, especially when it comes to a preclinical study that has interesting results but cannot make any clinical claims. The virtue of this manuscript is, not that it is without flaws ore weaknesses, but that it provides a background that helps our understanding of this technique in a manner that could not be possible in clinical studies. However, it cannot "contradict/challenge" published clinical trials published after the experiments were conducted. The knowledge has already moved forward. Please revise.
8. Regarding SLND and SLNB. My point is that some methods, such as, CEUS with microbubbles can obtain a SLNB without the need for SLND. Others seek to omit SLND by minimally invasive or non-invasive methods. I understand the argument that preclinical literature uses the term SLNB. But what you describe, and its potential for the clinical setting is SLND. So, please revise.
9. Please check the supplementary files. You have probably copy/pasted it from the statistical software and what the reader sees is only #####. This is very unfortunate. Please provide formatted tables in line with the Journal style.
In conclusion, the manuscript is much better and much closer to the mark but the aforementioned queries must be addressed and the manuscript revised accordingly. The revisions are not major, but the improvement will be.
Author Response
Response to Reviewer 1
The authors have addressed the majority of my comments in a satifactory manner and have provided a greatly improved manuscript. Some further points must be revised, however, so as to provide a consistent and relevant manuscript.
Thank you very much for your valuable comments. Your suggestions and comments have improved the manuscript significantly. We tried to carefully address the remaining points that you raised. We hope that the quality of the paper reached to the point required for publication.
- p2, r64-69: "However, none of these studies have had a comprehensive look at the optimal magnetic nanoparticles injection for SLNB, and questions remain regarding in body dynamics of the agent, to determine the optimum iron dose, optimum time course of injection, the efficacy of dilution and efficacy of massaging the injection site. These factors are important in order to reduce the residual magnetic nanoparticles left at the injection site.
The "comprehensive look" is a little bit unclear of a formulation. These are clinical studies and they had a different endpoint; it would probably not be possible to have ethical approval to study the body dynamics of SPIO on humans and this is why this paper has its role. Please rephrase, as it now seems that you are "questioning" the studies you are citing, whereas it is not so.
Thank you for your careful reviewing. As you commented, we did not try to question the results from the previous studies. Rather, we appreciated them as they suggested the factors to be optimized. We realized the previous sentence was inappropriate and revised the text as follows:
Introduction, lines 65-68
Knowledge from the previous studies indicated that iron dose, time course of injection, dilution, and massaging could affect iron uptake by the sentinel lymph nodes (SLN). Optimization of these factors are important in order to reduce the residual magnetic nanoparticles left at the injection site. - 3, r 108-109: "The caudal skin of the stifle joint was incised, and the popliteal lymph nodes were visually identified at its anatomical location".
In your response, you write that no probe was used to identify the nodes, if I understand correctly. This means that "visually identified" refers to color? Please clarify. Additionally, if you did not take ex vivo counts, please clarify as it is a limitation, given the fact that, for clinicians that would use a magnetic biopsy, it would be harder to correlate to amount of iron; instead they would think of probe counts and color, such as in the case of isotope and blue dye.
Thank you for your instruction. We are sorry that the previous writing was unclear. We meant that the lymph nodes were anatomically located between the thigh muscles at the stifle joint. In most cases, the SLNs are colored brown depending on the injection dose. However, with low dose injection, the SLN coloring was slight or indistinguishable from the normal node. And we understand that it is another limitation that we could not take magnetometer count ex vivo in this study. Accordingly, the text was revised as follows
Materials and methods, lines 105-110
In rats, the popliteal lymph nodes are the primary and dominant SLNs draining the distal hindleg including the injection site. Following euthanasia, SLNB of the popliteal nodes was performed bilaterally in a prone position. The caudal skin of the stifle joint was incised, and the popliteal lymph nodes were anatomically located within the thigh muscles and dissected (Figure 1a).
Results, lines 204-205
The higher dose injection of Resovist resulted in apparent brown discoloration of the SLNs (Figure 1a).
Discussion, lines 448-452
A major limitation of this study is that we used a rat model to optimize SPIO injection. Although it enabled us to address the factors associated with the injection, the SLNs of rats were small, and ex vivo counts by a magnetometer were not taken in this study. Therefore, the theoretical basis provided in this study should be further evaluated in more relevant animal models, which may include dog and pig, or in clinical studies. - p5, r 229-232. "A preliminary observation indicated that iron accumulation in the secondary nodes would increase with the iron injection dose as in SLNs (Figure 1d), although statistical testing was not performed due to the limited number of animals involved."
Since no statistical testing was performed, this needs to be removed completely.
Thank you for pointing out this part. We removed the phrases that were not supported by the statistics. The text was revised as follows:
Results, lines 229-231
The secondary nodes contained the similar amount of iron as in the SLNs at the investi-gated conditions (1,000-, 2,790-, and 4,000-μg injection) (Figure 1d). - p12, r456-458: "Preoperative injection more than one day, up to months, prior to surgery needs to be investigated in the future as well, to see if longer waiting period facilitates more accumulation of iron in the SLNs and/or leads to complicating accumulation in secondary and tertiary nodes. "
This is very relevant. Looking into the literature, there is this recent publication (Jazrawi et al, Cancers 2021, 13(17), 4285; https://doi.org/10.3390/cancers13174285). Here, some patients were injected with SPIO prior to neoadjuvant therapy and were explored with axillary MRI before and after. Then, surgery was performed after neoadjuvant with the coadministration of isotope. The authors found that the same nodes were visualized and detected by both methods. This makes the case that it probably does not migrate higher. Please discuss this.
Thank you for providing the latest information. I went through the manuscript and agreed that SPIO is not likely to migrate to subsequent lymph nodes weeks or even months after injection. Accordingly, we added several sentences to Discussion as follows:
Discussion, lines 456-460
In a recent study, patients were injected with SPIO prior to neoadjuvant chemotherapy, and MRI lymphography was compared before and after chemotherapy with a median of 130 days interval [33]. As a result, SPIO accumulation was observed in the same lymph nodes. Therefore, it was suggested that SPIO does not migrate in higher nodes for months, which would support SPIO injection more than 1 day prior to surgery. - 11, r.403-404: "Some articles looked at aspects of the SPIO injection, but no comprehensive clinical studies were performed."
Again, "comprehensive" is rather vague and the comment is inaccurate. The studies by Rubio and Hersi that you cite look into the effect of dose, injection site and injection site. So they are as comprensive as it can be, given that they are clinical trials. Please omit.
Thank you again for careful reviewing. We totally agreed with your opinion. We revised the text as follows:
Discussion, lines 401-402
Previous clinical studies also evaluated different protocols of the SPIO injection. - 13, r 468-471: "As such, peritumoral injection of SPIO would have different consequences from subcutaneous injection into the normal tissue performed in this study. Different lymphatic system, such as the axillary versus neck networks, may have different draining machineries."
How would it have consequences? Do we know it? Or is it an assumption? Please revise.
Thank you for pointing this out. I apologize for your confusion. I think my wording was misleading. We imagined that peritumoral lymphatic flow may be dysregulated and SPIO clearance would be affected as well, as Reviewer 2 suggested. We revised the text accordingly. I hope this version will not cause readers’ confusion.
Discussion, lines 469-471
As such, peritumorally injected SPIO would be put in different lymphatic dynamics com-pared to subcutaneous injection into the normal tissue performed in this study. - p 13, r.478-479: "Probable benefits from longer waiting period more than one day should be tested in the future."
You are already citing studies with demonstrated benefits from a much longer waiting period. So this does not need to be tested. Revise in a manner that contains this message. The experiments were conducted previously, but they are presented now, so the results need to placed in this context accurately. As with comment#5, the point is that the writing style in some points looks somewhat aggressive towards the cited literature, as if trying to deconstruct it, in order to show what has not been done yet and what needs to be done. It is my understanding that this has previously been a more popular writing style, but it is very important to be able to set things into scientific context. This means that claims need to be completely transparent, justified and evidence-based. I cannot stress how important this is for the value and the appropriateness of any manuscript, especially when it comes to a preclinical study that has interesting results but cannot make any clinical claims. The virtue of this manuscript is, not that it is without flaws ore weaknesses, but that it provides a background that helps our understanding of this technique in a manner that could not be possible in clinical studies. However, it cannot "contradict/challenge" published clinical trials published after the experiments were conducted. The knowledge has already moved forward. Please revise.
Thank you very much for your instruction and precious guidance regarding scientific writing. We totally agreed with your opinion, and it was our failure that the previous text seemed aggressive to the previous studies. We appreciate knowledge from them. Our study is totally dependent on the previous knowledge. We also agreed that preclinical studies cannot make any clinical claims but would provide a background for understanding. Accordingly, we revised the text as follows. We also checked the entire manuscript to make sure that readers would not get such impressions.
Conclusions, lines 478-482
From these results, preoperative injection may be advantageous for SLN detection, as has been shown in the pioneering clinical studies, as well as for reduction in residual iron at the injection site by expected decrease in required injection dose. The result from this study provides a theoretical background that helps our understanding of the magnetic SLND. - Regarding SLND and SLNB. My point is that some methods, such as, CEUS with microbubbles can obtain a SLNB without the need for SLND. Others seek to omit SLND by minimally invasive or non-invasive methods. I understand the argument that preclinical literature uses the term SLNB. But what you describe, and its potential for the clinical setting is SLND. So, please revise.
Thank you for your detailed explanation. After consideration, we agreed with your assessment. While SLNB can be different methods, what we describe, and its clinical counterpart is SLND. We revised the entire text accordingly. When it first appears, I introduced “sentinel lymph node biopsy” as well, for people serching for publications with a keyword of SLNB.
Introduction, lines 40-43
Sentinel lymph node dissection (SLND), also referred to as sentinel lymph node biopsy, is the standard of care for clinically and radiologically node-negative breast cancer patients to stage the axilla and determine if cancer has spread to regional lymph nodes [1,2].
- Please check the supplementary files. You have probably copy/pasted it from the statistical software and what the reader sees is only #####. This is very unfortunate. Please provide formatted tables in line with the Journal style.
Thank you again for your careful reviewing. We formatted the tables according to the model article. - In conclusion, the manuscript is much better and much closer to the mark but the aforementioned queries must be addressed and the manuscript revised accordingly. The revisions are not major, but the improvement will be.
We believe that your comments have improved the manuscript a lot. Thank you very much.
